# Quadrupolar magnetic excitations in an isotropic spin-1 antiferromagnet

A. Nag [1,7 ✉], A. Nocera[2,3 ✉], S. Agrestini[1], M. Garcia-Fernandez [1], A. C. Walters [1], Sang-Wook Cheong [4], S. Johnston [5,6 ✉] & Ke-Jin Zhou [1 ✉]

The microscopic origins of emergent behaviours in condensed matter systems are encoded in their excitations. In ordinary magnetic materials, single spin-flips give rise to collective dipolar magnetic excitations called magnons. Likewise, multiple spin-flips can give rise to multipolar magnetic excitations in magnetic materials with spin $S \geq 1$. Unfortunately, since most experimental probes are governed by dipolar selection rules, collective multipolar excitations have generally remained elusive. For instance, only dipolar magnetic excitations have been observed in isotropic $S = 1$ Haldane spin systems. Here, we unveil a hidden quadrupolar constituent of the spin dynamics in antiferromagnetic $S = 1$ Haldane chain material $Y_2BaNiO_5$ using Ni $L_3$-edge resonant inelastic x-ray scattering. Our results demonstrate that pure quadrupolar magnetic excitations can be probed without direct interactions with dipolar excitations or anisotropic perturbations. Originating from on-site double spin-flip processes, the quadrupolar magnetic excitations in $Y_2BaNiO_5$ show a remarkable dual nature of collective dispersion. While one component propagates as non-interacting entities, the other behaves as a bound quadrupolar magnetic wave. This result highlights the rich and largely unexplored physics of higher-order magnetic excitations.

[1] Diamond Light Source, Harwell Campus, Didcot OX11 0DE, United Kingdom. [2] Stewart Blusson Quantum Matter Institute, University of British Columbia, Vancouver, British Columbia V6T 1Z4, Canada. [3] Department of Physics Astronomy, University of British Columbia, Vancouver, British Columbia V6T 1Z1, Canada. [4] Rutgers Center for Emergent Materials and Department of Physics and Astronomy, Rutgers University, Piscataway, NJ, United States. [5] Department of Physics and Astronomy, The University of Tennessee, Knoxville, TN 37966, United States. [6] Institute for Advanced Materials and Manufacturing, University of Tennessee, Knoxville, TN 37996, United States. [7] Present address: SwissFEL, Paul Scherrer Institut, 5232 Villigen, Switzerland. ✉email: abhishek.nag@psi.ch; alberto.nocera@ubc.ca; sjohn145@utk.edu; kejin.zhou@diamond.ac.uk

The elementary excitation of a magnetically ordered material is a single dipolar spin-flip of an electron, delocalised coherently across the system in the form of a spin wave. The spin-wave quasiparticle, known as a magnon, carries a spin angular momentum of one unit and has well-defined experimental signatures. Collective dipolar magnetic excitations also appear in low-dimensional magnets that remain disordered to the lowest achievable temperatures because of quantum fluctuations. A paradigmatic example is the $S = 1$ antiferromagnetic Haldane spin chain, where magnetic order is suppressed in favour of a singlet ground state with nonlocal topological order[1,2]. Several theoretical and experimental works have established that a single spin-flip from this exotic ground state creates dipolar magnetic excitations that propagate along the chain above an energy gap of $\Delta_H \sim 0.41J$, the Haldane gap[3–8].

Materials hosting $3d$ transition metal ions with a $d^2$ or $d^8$ configuration (such as $Ni^{2+}$ for the latter) often possess strongly interacting spin-1 local magnetic moments with quantised spin projections $S_z = -1, 0, 1$. In addition to the usual single spin-flip excitations, it is possible to create quadrupolar magnetic excitations by changing the composite spin by two units. Such excitations can be conceived as flipping two of the constituent spin-1/2's, as shown in Fig. 1a. Incidentally, quadrupolar magnetic waves arising from such transitions and carrying two units of angular momentum were predicted for $S = 1$ ferromagnetic chains as early as the 1970s[9,10] and may play a role in the iron pnictide superconductors[11]. Since most probes are restricted by dipolar selection rules, however, such quadrupolar excitations have largely evaded detection except in rare situations where they are perturbed by anisotropic interactions, spin-orbit coupling, lattice vibrations, or large magnetic fields[12–17]. Quadrupolar magnetic excitations have never been observed in isotropic Haldane spin chains, even though dipolar magnetic excitations have been extensively studied[3–8,18]. It is then natural to wonder whether purely quadrupolar collective magnetic excitations exist in isotropic spin-1 systems.

Here, we uncover the presence of collective quadrupolar magnetic excitations in the *isotropic* $S = 1$ Haldane chain system $Y_2BaNiO_5$ using high energy-resolution Ni $L_3$-edge resonant inelastic x-ray scattering (RIXS). Previous studies on nickelates have already shown that Ni $L_3$-edge RIXS can probe dipolar magnons[19,20]. In addition, ref. [21] recently showed that double spin-flips are allowed at this edge through the combined many-body effect of core-valence exchange and core-hole spin-orbit interactions[22,23] (Fig. 1b), making it the optimal tool for this study. $Y_2BaNiO_5$ is one of the best realisations of the isotropic Haldane spin chain material with intra-chain exchange $J \sim 24$ meV and negligibly small single-ion anisotropy $\sim 0.035J$, exchange-anisotropy $\sim 0.011J$, and inter-chain exchange $\sim 0.0005J$[6,24]. This aspect allows us to describe the system completely using a simple Heisenberg model, and emphasises the relevance of the pure quadrupolar magnetic excitations in the spin dynamics of $S = 1$ systems.

## Results

**Excitations in Haldane spin chains.** In a valence bond singlet scheme, the Heisenberg model's ground state in the Haldane phase can be represented as a macroscopic $S_{tot} = 0$ state comprised of pairs of fictitious spin-$\frac{1}{2}$ particles on neighbouring sites that form antisymmetric singlets on each bond[8] (Fig. 1c). A single local spin-flip breaks a bond singlet to form a triplet excitation, raising the chain's total spin quantum number to $S_{tot} = 1$. In contrast, a local double spin-flip would disrupt singlets on either side of the excited site, creating a pair of triplet excitations and raising the spin quantum number to $S_{tot} = 2$. In the RIXS process, it is also possible to have a total spin-conserved two-site excitation with $\Delta S_{tot} = 0$ (see Fig. 1c), which appears as a two-magnon

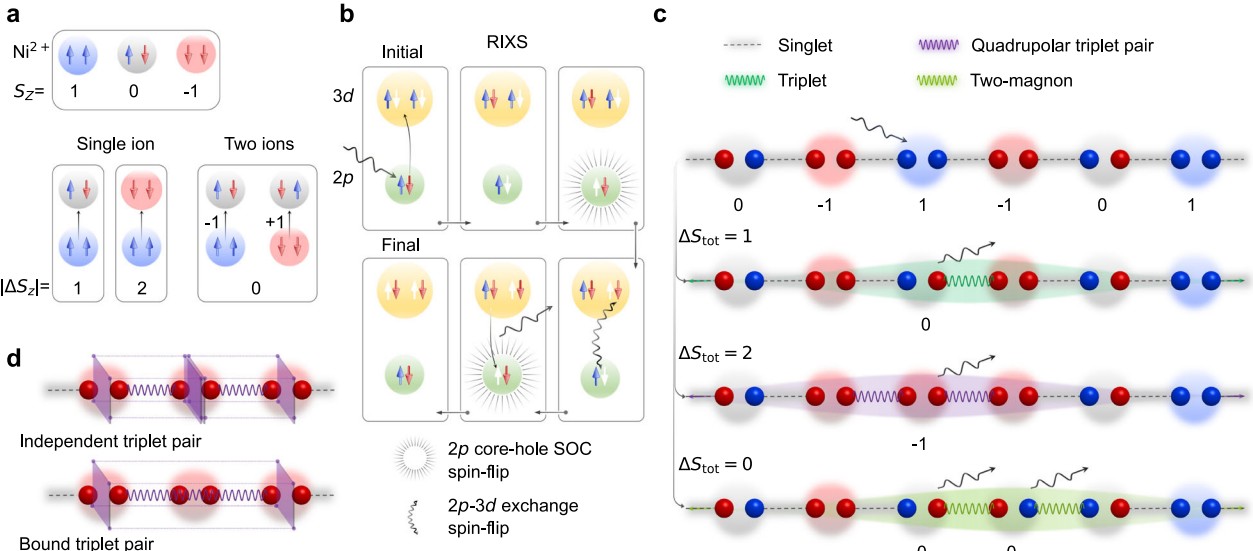

**Fig. 1 Magnetic excitations in S = 1 Haldane chains. a** Possible unpaired spin configurations in an $S = 1$ ion ($Ni^{2+}$) and spin-flip transitions. **b** Process of probing single-site quadrupolar $\Delta S_{tot} = 2$ excitations in $L$-edge RIXS. An $L$-edge RIXS process of a $S = 1$ $3d^8$ ion (with two unpaired $d$ electrons and ground state having $S_{tot} = 0$) proceeds via creation of an intermediate $2p^5 3d^9$ state. In this intermediate state, due to the strong spin-orbit coupling of $2p^5$ core-hole, the spin-angular momentum is not conserved leading to a valence orbital spin-flipped final state with $S_{tot} = 1$. Additionally, due to the many-body core-valence Coulomb exchange interactions in the intermediate states, an additional available valence orbital spin-flip may occur thereby creating an excited final state with $S_{tot} = 2$[21]. The coloured (white) arrows represent the occupied (empty) spin states of the electrons involved in the scattering process. **c** A representative Haldane spin chain characterised by the topological non-local (string) order of alternating $S_z = \pm 1$ sites intervened by any number of $S_z = 0$ sites. Between neighbouring sites a pair of spin-1/2s form an antisymmetric singlet. Single-site single spin-flips give rise to dipolar singlet-triplet excitations while single-site double spin-flips give rise to quadrupolar excitations resulting in pairs of triplets. Single spin-flips at multiple sites give rise to two-magnons. **d** Schematic representation of dual nature of propagation of the pair of triplets formed after quadrupolar excitation.

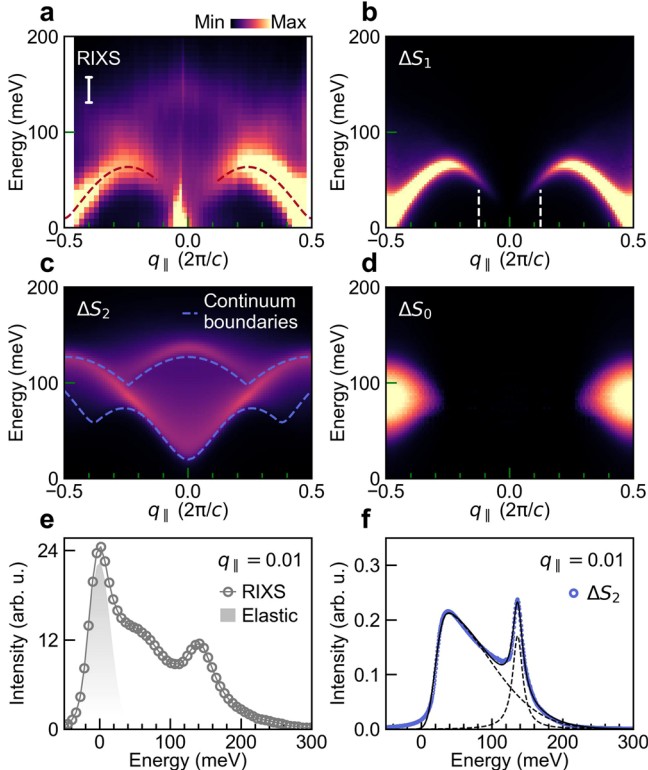

**Fig. 2 RIXS results and DMRG calculations for Y₂BaNiO₅. a** Experimental RIXS intensity map at Ni $L_3$-edge at 11 K. Dashed line is a semiquantitative dispersion $\omega^2(q_{\parallel}) = \Delta_H^2 + v^2\sin^2 q_{\parallel} + \alpha^2\cos^2\frac{q_{\parallel}}{2}$ with $J = 24$ meV, $\Delta_H = 0.41J$, $v = 2.55J$ and $\alpha = 1.1J$[4-6,49]. The vertical bar shows the experimental energy resolution. DMRG calculated dynamic spin susceptibility intensity maps for the **b**, $\Delta S_1$, **c**, $\Delta S_2$ and **d**, $\Delta S_O$ excitations (see Methods for details). In panel **c** we show the continuum boundaries expected for two independent triplet excitations (equivalent to two-magnon continuum boundaries). **e** RIXS line profile at $q_{\parallel} = 0.01$ and a shaded elastic peak profile. **f** Calculated profile for $\Delta S_2$ at $q_{\parallel} = 0.01$ from DMRG. The $\Delta S_2$ spectral weight can be decomposed into a broad continuum and a sharp peak (see Methods).

continuum. To simplify our notation, we will refer to the single-site single-spin-flip induced dipolar $\Delta S_{tot} = 1$ excitation as $\Delta S_1$, the single-site double-spin-flip induced quadrupolar $\Delta S_{tot} = 2$ excitation as $\Delta S_2$, and the two-site $\Delta S_{tot} = 0$ two-magnon excitation as $\Delta S_0$.

Figure 2a shows a RIXS intensity map collected on Y₂BaNiO₅ (see Methods). The feature with the largest spectral weight follows a dispersion relation consistent with inelastic neutron scattering (INS) results for the dipolar $\Delta S_1$ excitation[4,5,24]. This feature is also reproduced in our density matrix renormalisation group (DMRG) calculations of the dynamical structure factor $S_1(q_{\parallel}, \omega)$ for an isotropic Heisenberg model (see Fig. 2b and Methods). Ni $L_3$-edge RIXS is unable to reach the exact antiferromagnetic zone center ($q_{\parallel} = 0.5$, in units of $2\pi/c$ throughout), where the Haldane gap of ~ 8.5 meV exists. But it does probe an equally interesting region close to $q_{\parallel} = 0$. Prior work[4,7,25] has focused on observing the breakdown of the well-defined $\Delta S_1$ quasiparticle into a two-magnon continuum for $q_{\parallel} \lesssim 0.12$ and the spectral weight vanishing as $q_{\parallel}^2$. Although we notice a reduction in the intensity of the $\Delta S_1$ excitation in our experiment, we also observe significant inelastic spectral weight near zero energy close to $q_{\parallel} = 0$ (also see Fig. 2e for the line spectrum at $q_{\parallel} = 0.01$). Interestingly, a new dispersing excitation is clearly visible with an energy maximum of ~ 136 meV at $q_{\parallel} = 0$. Of the calculated

$S_0(q_{\parallel}, \omega)$, $S_1(q_{\parallel}, \omega)$, $S_2(q_{\parallel}, \omega)$ dynamical structure factors, shown in Fig. 2d, b, c, respectively, only $S_2(q_{\parallel}, \omega)$ has spectral weight in this region. Moreover, the $S_2(q_{\parallel} \sim 0, \omega)$ line profile has two components: a broad low-energy continuum and a sharp high-energy peak, as shown in Fig. 2f. It therefore appears that both the low- and high-energy spectral components seen in the experiment can be described by the quadrupolar $\Delta S_2$ excitation in this region of momentum space.

To evaluate the relative contributions of each excitation, we decomposed the RIXS spectra across the momentum space into the dynamical structure factors calculated by DMRG. Figure 3a–c show representative RIXS spectra. The data are fitted with an elastic peak and experimental energy-resolution convoluted line profiles of $S_\alpha(q_{\parallel}, \omega)$ ($\alpha = 0, 1, 2$, see Methods). While the spectra are dominated by the dipolar $\Delta S_1$ excitations at high $q_{\parallel}$ values, the spectral weight close to $q_{\parallel} = 0$ can be fitted with only the quadrupolar $\Delta S_2$ excitation. Figure 3d and e show the RIXS spectra after subtracting other fitted components to keep only the $\Delta S_1$ and $\Delta S_2$ contributions, respectively, along with their fitted profiles from DMRG. This analysis shows that the low-energy RIXS spectral weight is carried by the low-energy continuum component of the $\Delta S_2$ excitations below the quasiparticle decay threshold momentum. This component is gapped and has a peak energy of ~19 meV at $q_{\parallel} \sim 0$, reminiscent of the lower boundary of the two-magnon continuum at $2\Delta_H$ predicted for Haldane spin chains[7]. We note that the energy gap at this momentum has not yet been confirmed for any Haldane spin chain by INS due to the small scattering cross-sections. Figure 3f shows the RIXS intensity map after subtracting only the elastic peaks and Fig. 3g shows the combined DMRG dynamical structure factors $S_0(q_{\parallel}, \omega)$, $S_1(q_{\parallel}, \omega)$, and $S_2(q_{\parallel}, \omega)$ obtained by fitting the RIXS spectra. The $\Delta S_0$ type of two-magnon continuum excitations provide a negligible contribution to the RIXS spectra (see Methods for the contributions from each type of excitation). Undoubtedly, the two-component excitation (the broad low-energy continuum and the sharp high-energy component), and its dispersion in the momentum space (Fig. 3e) is well described only by the quadrupolar $\Delta S_2$ excitation.

**Quadrupolar excitations at finite $T$**. To further understand the character of the two-component $\Delta S_2$ quadrupolar excitations, we also studied their thermal evolution. Figure 4a shows the RIXS spectra at $q_{\parallel} = 0.01$, with contributions from the $\Delta S_2$ excitations, for increasing temperatures. For comparison, Fig. 4b shows finite temperature DMRG simulations for the $S_2(q_{\parallel} = 0.01, \omega)$ excitations, convoluted with the experimental energy resolution. Since the raw DMRG data for the $\Delta S_2$ channel (Fig. 2f) contains a two-peak structure with an asymmetric peak appearing at low energy and a symmetric peak at high energy, we fit the experimental $S_2(q_{\parallel} = 0.01, \omega)$ finite temperature data using two components (see methods). Our results show that only the $\Delta S_2$ excitations obtained from DMRG are needed to reproduce the experimental RIXS spectra, even at finite temperatures. As shown in Fig. 4c, the peak energy of the lower component increases with temperatures following twice of the system's Haldane gap from ref. [5]. Conversely, the high-energy peak begins to soften above the Haldane gap temperature of ~ 100 K. The highest energy value of the $\Delta S_1$ peak at $q_{\parallel} = 0.25$ also follows this trend. The bandwidth reduction of the $\Delta S_1$ triplet dispersion with temperature occurs due to the thermal blocking of propagation lengths and decoherence[26]. In a simple picture, if one considers a continuum from pairs of non-interacting triplets due to single spin-flips at multiple sites, then bandwidth reduction of each would manifest as the overall raising and lowering of the lower and upper boundaries of the continuum, respectively (see Supplementary Note 3). A similar thermal effect on the propagation of the quadrupolar $\Delta S_2$

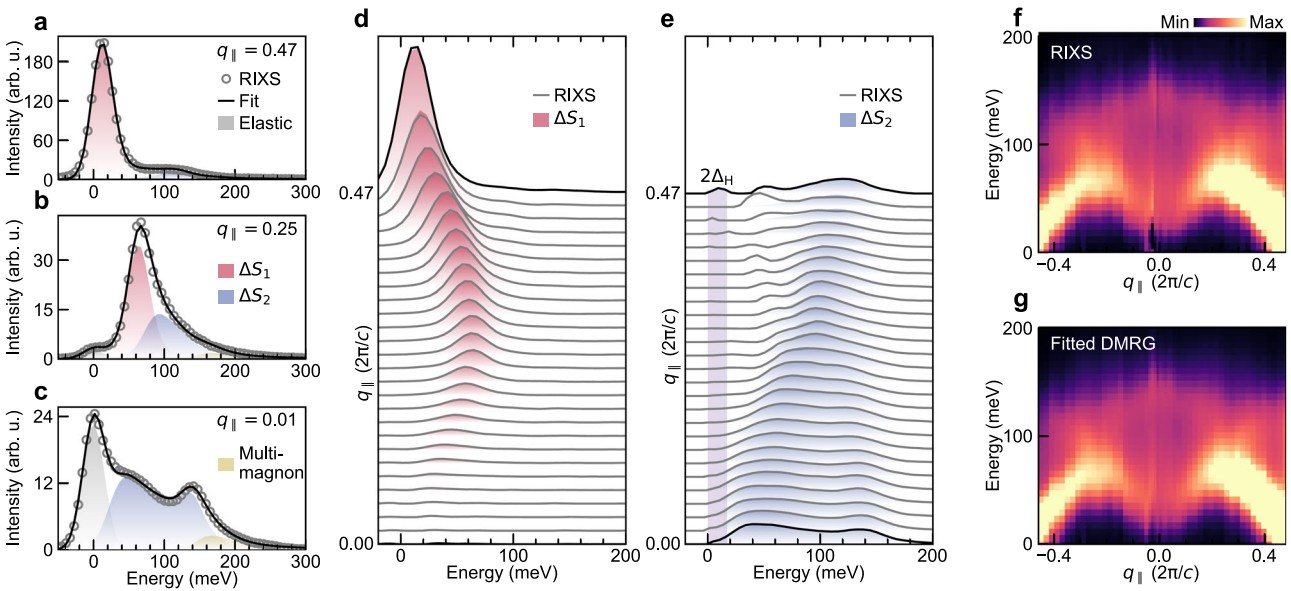

**Fig. 3 Magnetic excitations probed by RIXS in $Y_2BaNiO_5$ at Ni $L_3$-edge.** Representative RIXS spectra at $q_\parallel = $ **a**, 0.47, **b**, 0.25, and **c**, 0.01 with fits using spin susceptibilities obtained from DMRG (see Methods). Note that an additional peak is included in the fits in panel **c** to account for a multimagnon spectral tail around 200 meV, which may originate from higher-order contributions not considered in the expansion of the RIXS response. **d** RIXS line spectra after subtraction of elastic, $\Delta S_2$ and the high energy tail structure visible in panel **c**. Shaded areas are fitted profiles from DMRG for $\Delta S_1$. **e** RIXS line spectra after subtraction of elastic, $\Delta S_1$ and the high energy tail structure visible in panel **c**. Shaded areas are fitted profiles from DMRG for $\Delta S_2$. The vertical shaded bar represents twice the $\Delta_H$ value from ref. [5]. **f** Experimental RIXS intensity map with elastic contribution subtracted. **g** DMRG intensity map with summed contributions from $\Delta S_0$, $\Delta S_1$ and $\Delta S_2$ excitations determined by fitting experimental RIXS spectra using Eq. (5).

excitation should occur. The spectral weight of the two components in the $\Delta S_2$ excitation, however, behave differently with temperature. The low-energy continuum intensity varies little, while the high-energy peak diminishes rapidly with increasing temperature and the rate of decay is comparable to the $\Delta S_1$ peak at $q_\parallel = 0.47$ (see Fig. 4d). The DMRG calculated correlation lengths (in lattice units) of the $\Delta S_1$ and $\Delta S_2$ excitations, as shown in Fig. 4e, also decrease in a similar way with temperature.

**Dual nature of the quadrupolar excitations**. The energy-momentum and temperature dependence of the quadrupolar $\Delta S_2$ excitations implies that their low- and high-energy components are different in nature. At low-energy, a conceivable picture is that immediately after a pair of triplets are created by a single-site excitation, they decay into two noninteracting triplets (see Fig. 1d) propagating incoherently along the chain and giving rise to the broad low-energy continuum (see Fig. 2c for the expected continuum boundaries). This behaviour is remarkably similar to the fractionalisation of a $\Delta S_{tot} = 1$ excitation into a two-spinon continuum in isotropic spin-1/2 chains[27], but having a distinct origin.

The sharpness of the high-energy component and its rapid decay with temperature, on the other hand, hints that it behaves as a distinct quasiparticle formed from pairs of triplets propagating coherently (see Fig. 1d)[28,29]. In low-dimensional systems, sharp peaks in the magnetic spectrum may either originate from a van Hove singularity in the density of states of quasiparticles (in our case, non-interacting triplet pairs) or from the formation of a bound state (in our case, bound triplet pairs). In Fig. 2c, the lower and upper boundaries of the continuum from pairs of non-interacting triplets (equivalent to the two-magnon continuum) are shown. The high-energy component of $\Delta S_2$ appears above the upper boundary of the continuum, ruling out the van Hove singularity scenario and suggesting the formation of

a bound state. The peak energy of the high-energy component is slightly larger than twice the highest energy value of the $\Delta S_1$ peak (by ~ 7 meV at $T = 11$ K) and, surprisingly, remains so up to the highest measured temperature. The small positive energy difference suggests a weak *repulsive* interaction between the bound triplets formed after a quadrupolar $\Delta S_2$ excitation[9,30]. Supplementary Note 6 provides a semi-quantitative energy scale based argument to support the notion of the bound state of weakly repulsing triplets excitations.

The correlation length of the $\Delta S_1$ excitations in Haldane spin chains decay exponentially in the presence of a non-local order, which can be viewed to originate from alternating $S_z = \pm 1$ sites intervened by $S_z = 0$ sites. The $\Delta S_2$ excitations also have an exponentially decaying correlation length, albeit smaller than $\Delta S_1$ excitations. The difference in correlation length is likely due to the fact that the creation of quadrupolar wave excitations costs energetically at least twice as much the single spin-flip magnon-like excitations at any wave-vector. As such, they may provide a means to detect the hidden non-local order that, at present, is only estimated theoretically by considering the $S_z = \pm 1$ states in Haldane spin chains[2,31,32]. Overall, we show that the $\Delta S_2$ excitations sustain the isotropic properties, the exchange interactions, and the coherence inherent to the system.

## Discussion and outlook

Magnetic excitations provide vital information about a system's thermodynamic, magneto-transport, ultrafast magnetic, spintronic, or superconducting properties. Moreover, higher-order multipolar spin and orbital degrees of freedom give rise to exotic non-classical phenomena like the Kitaev spin liquid[33], multispinons[34], spin-nematicity[15], on-site multiferroicity[12], and bound-magnon states [14,16], in a wide variety of magnetic systems. However, as noted earlier, multipolar excitations are challenging to detect using conventional probes[35]. It was shown recently that

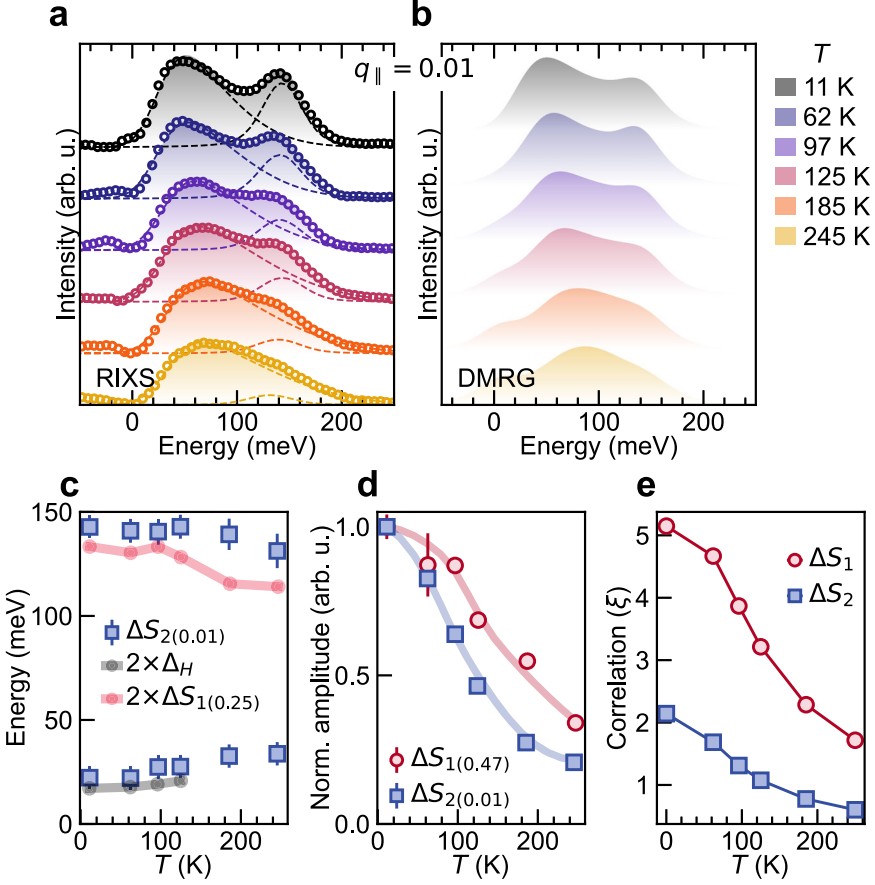

**Fig. 4 Thermal stability and correlations of quadrupolar excitations in $Y_2BaNiO_5$. a** RIXS spectra (open circles) at $q_\parallel = 0.01$ with fitted elastic contributions subtracted, at different temperatures. The $\Delta S_2$ contributions have been fitted to extract the positions and amplitudes of the two branches shown by dashed lines (see Methods). The shaded regions show the complete fit profiles. **b** $\Delta S_2$ profiles obtained from DMRG for corresponding temperatures. **c** Energy variations with temperature of the two branches of the $\Delta S_2$ excitations extracted from RIXS shown in panel **a**. Blackline with circles shows the thermal variation of twice the Haldane gap energy obtained from ref. [5]. Redline with circles shows the thermal variation of twice the $\Delta S_1$ peak energy at $q_\parallel = 0.25$. **d** Thermal variation in amplitudes of $\Delta S_1$ peak and the high energy peak of the $\Delta S_2$ extracted from fitted RIXS spectra at $q_\parallel = 0.47$ and 0.01, respectively, normalised to corresponding amplitudes at 11 K. **e** Dipolar and quadrupolar correlation lengths for $\Delta S_{tot} = 1$ and 2 excitations obtained from DMRG static dipolar and quadrupolar correlation functions (see Methods). Error bars are least-square-fit errors.

quadrupolar excitations in $S = 1$ $FeI_2$ appear in INS only due to their hybridisation with dipolar excitations through anisotropic spin-exchanges[14]. In the presence of a strong anisotropy, bound magnetic excitations from $\Delta S_{tot} = 2$ spin-flips at energies higher than the two-magnon continuum have also been observed in $S = 1$ spin chains using high-field electron spin resonance[36,37]. However, our demonstration of pure quadrupolar spin dynamics in an isotropic Haldane system, without invoking anisotropic interactions, suggests that simultaneous confinement and propagation of excitations can occur entirely via higher-order quantum correlations[38]. This work thus illustrates RIXS's capability of detecting higher-order dispersing excitations, irrespective of the presence of dipolar excitations[14], and thus may be the preferable way to study quadrupolar excitations in, for instance, spin-nematic systems, where the dipolar excitations are suppressed[39,40]. Also recently, it has been seen that Cu $L_3$-edge RIXS can probe spin-conserved and non-spin-conserved higher-order four-spinons in spin-1/2 Heisenberg antiferromagnetic spin chains[41]. Our present work further consolidates $L$-edge RIXS (in comparison to O $K$-edge)[34], as a powerful probe for characterising nonlocal long-range magnetic correlations via the study of higher-order spin-flip excitations, thereby, extending both the energy-momentum phase space and the diversity of the magnetic quantum systems that can be explored. On the other hand,

exploring the physics of high-energy excitations and/or eigenstates of a simple *nonintegrable* spin chain model is in itself of great theoretical interest. We provide a simple physical intuition about the nature of one of the high-energy eigenstates of the one-dimensional Heisenberg model and our findings may have important consequences for many-body correlated states of matter and thermalisation in quantum systems[42]. Looking forward, it would be interesting to learn how the quadrupolar excitations can be manipulated with intrinsic perturbations like anisotropy or extrinsic ones like a magnetic field. The indications of a propagating quadrupolar *bound* excitation in a real material can also have important ramifications for realising quantum information transfer in form of qubit pairs[43,44].

## Methods

**Experiments**. A single crystal of $Y_2BaNiO_5$ grown by the floating-zone method was used for the RIXS measurements. The momentum transfer along the chain direction $q_\parallel$ was varied by changing the x-ray incident angle $\theta$ while keeping the scattering angle fixed at 154°. The lattice constant along the chain or the $c$-axis used for the calculation of momemtum transfer is 3.77 Å. The crystal was cleaved in vacuum and the pressure in the experimental chamber was maintained below $\sim 5 \times 10^{-10}$ mbar throughout the experiment. High energy-resolution RIXS data ($\Delta E \simeq 37$ meV) at the Ni $L_3$-edge were collected at I21 RIXS beamline, Diamond Light Source, United Kingdom[45]. The zero-energy position and resolution of the RIXS spectra were determined from subsequent measurements of elastic peaks from an adjacent carbon tape. The polarization vector of the incident x-ray was

parallel to the scattering plane (i.e. $\pi$ polarization). See Supplementary Note 1 for more details of the experimental configuration.

**Theory**. In the main text, we pointed out that the $S = 1$ Haldane chain system $Y_2BaNiO_5$ might present negligibly small single-ion anisotropy ~$0.035J$, exchange-anisotropy ~$0.011J$ terms in a Heisenberg model description at low energies. We have verified numerically that these small corrections do not change our magnetic spectra qualitatively, and therefore a pure isotropic Heisenberg model has been adopted throughout our study.

*Zero and finite temperature DMRG Calculations.* $T = 0$ DMRG calculations on 100 site chains with open boundary conditions (OBC) were carried out with the correction-vector method[46] using the Krylov decomposition[47], as implemented in the DMRG++ code[48]. This approach requires real-space representation for the dynamical structure factors in the frequency domain, which can be found in the Supplementary Note 4. For $T > 0$ calculations we used the ancilla (or purification) method with a system of 32 physical and 32 ancilla sites, also with OBC. For more details see Supplementary Note 7.

For both the zero temperature and finite temperature calculations we kept up to $m = 2000$ DMRG states to maintain a truncation error below $10^{-7}$ and $10^{-6}$, respectively and introduced a spectral broadening in the correction-vector approach fixed at $\eta = 0.25J = 6$ meV.

*Dynamical spin correlation functions.* We consider three correlation functions $S_0(q_\parallel, \omega)$, $S_1(q_\parallel, \omega)$, and $S_2(q_\parallel, \omega)$ giving information about $\Delta S_{tot} = 0$, $\Delta S_{tot} = 1$, and $\Delta S_{tot} = 2$ excitations, respectively. To make the expressions more transparent, we use the Lehmann representation and construct the corresponding excitation operators in momentum space. The relevant correlation functions are

$$S_\alpha(q_\parallel, \omega) = \sum_f |\langle f|S^\alpha_{q_\parallel}|\psi\rangle|^2 \delta(\omega - E_f + E_\psi), \quad (1)$$

where $|f\rangle$ are the final states of the RIXS process and

$$S^0_{q_\parallel} = \frac{1}{\sqrt{L}}\sum_j e^{iq_\parallel j} \vec{S}_j \cdot \vec{S}_{j+1}, \quad (2)$$

$$S^1_{q_\parallel} = \frac{1}{\sqrt{L}}\sum_j e^{iq_\parallel j} S^+_j, \quad (3)$$

$$S^2_{q_\parallel} = \frac{1}{\sqrt{L}}\sum_j e^{iq_\parallel j} (S^+_j)^2. \quad (4)$$

The three dynamical correlation functions given by Eq. (1) appear at the lowest order of a ultrashort core-hole lifetime expansion of the full RIXS cross-section. As the single-site $\Delta S_{tot} = 0$ RIXS scattering operator is trivial (identity operator) in a low-energy description in terms of spin $S = 1$ sites, the lowest order operator would involve two-sites and, by rotational symmetry, involves a scalar product of neighbouring spin operators [see Eq. (2)]. Single and double spin-flip RIXS scattering operators, on the other hand, lead to $\Delta S_{tot} = 1$ and $\Delta S_{tot} = 2$ excitations and can be naturally described in terms of onsite $S^+_j$ and $(S^+_j)^2$ operators, respectively [Eqs. (3) & (4)]. In the Supplementary Note 5 we provide analysis of the three dynamical correlation functions in terms of single triplet excitations or *magnon* states in the Haldane chain.

*Correlation lengths.* Figure 4e of the main text shows dipolar and quadrupolar correlation lengths as a function of temperature. These have been obtained by computing $\langle\psi(\beta)|S^+_j S^-_{j+r}|\psi(\beta)\rangle$ and $\langle\psi(\beta)|(S^+_j)^2(S^-_{j+r})^2|\psi(\beta)\rangle$ correlation functions from the center of the chain $j = c$, respectively, and fitting with a exponential decay relationship $f(r) = Ae^{-r/\xi}$.

**RIXS data fitting**. RIXS data were normalised to the incident photon flux and corrected for x-ray self-absorption effects prior to fitting. The elastic peak was fit with a Gaussian function with a width set by the energy resolution. The RIXS spin excitations in Fig. 3 were modeled with the Bose factor weighted dynamical spin susceptibilities obtained from our $T = 0$ DMRG calculations for $\Delta S_{tot} = 0, 1,$ and 2 ($S_0, S_1$ and $S_2$, respectively) after they were convoluted with a Gaussian function capturing the experimental energy resolution. The total model intensity was given by

$$I_{RIXS}(q_\parallel, \omega) = C_0(q_\parallel)S_0(q_\parallel, \omega) + C_1(q_\parallel)S_1(q_\parallel, \omega) + C_2(q_\parallel)S_2(q_\parallel, \omega), \quad (5)$$

where the coefficients $C_0$, $C_1$ and $C_2$ account for the varying RIXS scattering cross section for each spin excitation with varying $\theta$ ($q_\parallel$). The reader is referred to the Supplementary Note 2 for the extracted values of coefficients and the fit profiles. As seen in Fig. 3c, $S_2$ from DMRG for $\Delta S_{tot} = 2$ transitions do not capture the additional spectral weight on the high energy side in the RIXS signal. A 'half'-Lorentzian truncated damped harmonic oscillator (HLDHO) function centered at 0.175 eV was therefore included in the fits to account for this tail. In Fig. 4, the $\Delta S_{tot} = 2$ excitations at $q_\parallel = 0.01(2\pi/c)$ are decomposed using a DHO function for the sharp high energy peak and a skewed Gaussian function for the broad lower continuum. Both functions are energy-resolution convoluted and weighted by a

Bose factor. DHO functions were used similarly for fitting $\Delta S_{tot} = 1$ triplet excitations at $q_\parallel = 0.47$ (see Supplementary Note 2).

## Data availability

The instruction to build the input scripts for the DMRG++ package to reproduce our results can be found in the Supplementary Material. The data to reproduce our figures is available as a public data set at https://doi.org/10.5281/zenodo.6394852. Raw data files will be made available upon request.

## Code availability

The numerical results reported in this work were obtained with DMRG++ versions 6.01 and PsimagLite versions 3.01. The DMRG++ computer program[48] is available at https://github.com/g1257/dmrgpp.git (see Supplementary Note 7 for more details.)

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

## Acknowledgements

We thank I. Affleck, J.-G. Park, S. Hayden, A. Aligia, and K. Wohlfeld for insightful discussions. We are grateful to C. Bastista for suggesting the possibility of a van Hove singularity in our results. All data were taken at the I21 RIXS beam line of Diamond Light Source (United Kingdom) using the RIXS spectrometer designed, built, and owned by Diamond Light Source. We acknowledge Diamond Light Source for providing the beam time on beam line I21 under Proposal MM24593. S. J. acknowledges support from the National Science Foundation under Grant No. DMR-1842056. A. Nocera acknowledges support from the Max Planck-UBC-UTokyo Center for Quantum Materials and Canada First Research Excellence Fund (CFREF) Quantum Materials and Future Technologies Program of the Stewart Blusson Quantum Matter Institute (SBQMI), and the Natural Sciences and Engineering Research Council of Canada (NSERC). S.-W.C. was supported by the DOE under Grant No. DOE: DE-FG02-07ER46382. This work used computational resources and services provided by Compute Canada and Advanced Research Computing at the University of British Columbia. We acknowledge T. Rice for the technical support throughout the beam times. We also thank G. B. G. Stenning and D. W. Nye for help on the Laue instrument in the Materials Characterisation Laboratory at the ISIS Neutron and Muon Source.

## Author contributions

K.-J.Z. conceived the project; K.-J.Z., A. Nag, A. Nocera, and S.J. supervised the project. A. Nag, K.-J.Z., S.A., M.G.-F., and A.C.W. performed RIXS measurements. A. Nag, S.A., and K.-J.Z. analysed RIXS data. S.-W.C. synthesized and characterised the sample. A. Nocera and S.J. performed DMRG calculations. A. Nag, K.-J.Z., A. Nocera, and S.J. wrote the manuscript with comments from all the authors.

## Competing interests

The authors declare no competing interests.
