## [Peer Review File · Nature Communications]

Reviewers' Comments:

Reviewer #1:

Remarks to the Author:

In the manuscript "Quadrupolar magnetic excitations in an isotropic spin-1 antiferromagnet" A. Nag et al. reveal the presence of quadrupolar magnetic excitations in the $S=1$ $Y_{2}BaNiO_{5}$ Haldane compound by means of the state of the art of Resonant Inelastic X-ray Scattering (RIXS) experiments. The high quality data presented in the manuscript are interpreted in terms of an isotropic Heisenberg model, whose dynamical structure factor is calculated by means of the density matrix renormalization group (DMRG) technique. The authors provide a careful characterization of the spectral features corresponding to the quadrupolar $\Delta S=2$ excitations by measuring their variation as a function of momentum transfer and temperature. In this way they unveil the presence of both coherently and incoherently propagating triplet pairs in the system.

I find the present manuscript properly written and well organized. The quality of the data is impressive and represents the state of the art of RIXS experiments. I find the interpretation of the spectral features corresponding to quadrupolar excitations and the comparison with the theoretical model convincing. However, I do not find surprising that RIXS can probe quadrupolar excitations. As presented by A. Nag et al. in Ref. 21 single site quadrupolar excitations were already measured in NiO. More interesting in my opinion is the dynamics of such excitations revealed in a compound such as the $S=1$ Haldane chain material as $Y_{2}BaNiO_{5}$. The importance of such a result with respect to open problems in magnetism and, more generally, strongly correlated systems is not stressed enough in the present manuscript. In my opinion, in order to meet the broad interest required by Nature communications the authors must state more clearly and more precisely the general interest of their results.

Reviewer #3:

Remarks to the Author:

Excitations in magnetic materials that involve a single spin-flip are easy to observe by standard methods, like neutron scattering or ESR. These are well-known magnons (sometimes triplons), and their dispersion provides valuable information on the properties of the phase and on the strength of interactions. They are also the Goldstone-modes of the ordered magnets with broken time-reversal symmetry.

Magnetic systems in the quantum limit may realize phases that are beyond the usual picture of ordered magnetic moments. The authors present an experimental and numerical study of the quadrupolar excitations in a quasi-one-dimensional material $Y_{2}BaNiO_{5}$. In this material, the $S=1$ spins of the Ni ions form a quantum AKLT-state with a singlet ground state separated by a gap from the excitations. The resonant inelastic X-ray scattering (RIXS) excites two spin-flip modes - these are the so-called quadrupolar excitations. They are extremely difficult to create and observe, as the experimental probes usually do not interact with them, with the exception of the light. In the actual experiment, the two-spin quadrupolar excitations decay into a continuum of two triplet excitations and a quadrupolar collective mode at higher energies, which is an anti-bound state of two triplets. The authors compare in detail the RIXS spectrum with numerical calculation of the $S=1$ spin chain and identify the origin of the different parts of the spectrum. In my opinion, this is very nice work and I would like to recommend the acceptance of the manuscript for publication. The presented experimental technique may become essential to provide experimental identification of nematic (quadrupolar) ordering of spins, suggested for some materials, by observing their Goldstone modes.

However, before publication I would like to make a few suggestions, mostly esthetic ones:

(1) The process depicted in Fig. 1(b) is essential. However, it is very difficult to comprehend what is precisely happening. I think the initial state is missing - would it be two spins on 3d orbitals and 2 spins on 2p orbitals? And I am confused with the number of white and colored arrows. Could the authors spend some time and improve the figure?

(2) Fig. 2 shows the measured spectra. The ticks are difficult to discern. Could they color ticks

differently (say green?). Also, I was trying to extend the figure to $q_{\parallel} = 0.5$, to the zone boundary. I think it would help to show where is the boundary of the (1D) Brillouin zone, and to show the fitted dispersion (dashed line) in the whole zone. Clearly, there will be some white space where there is no RIXS signal, but that would also be informative.

(3) I am confused with the sentence (line 131 of the manuscript): "because the quadrupolar excitations can only occur at the $S_z = \pm 1$ sites and not the $S_z = 0$ sites." Just like magnons can be excited with any of the S^+_x , S^+_z , and S^-_x operators, there is a $(S^z)^2 - 2/3$ quadrupolar operator which will act on the $S_z = 0$ sites (there are five quadrupolar operators, as they are rank-2 tensor operators, see e.g. the chapter about spin operators in the Messiah's textbook on Quantum Mechanics). Could the authors improve that part of the text (I understand that there is a conflict between simplicity and exactness, but as the sentence now reads, it may be misleading)

(4) I think the Supplemental Material will be printed as it is provided. If I am not mistaken, all the formulas were typeset using Word. It was really painful to read them - how about converting those parts of the text to latex (or anything that looks reasonable)?

Report of the First Referee

The Referee wrote: In the manuscript “Quadrupolar magnetic excitations in an isotropic spin-1 antiferromagnet” A. Nag et al. reveal the presence of quadrupolar magnetic excitations in the $S = 1$ Y_2BaNiO_5 Haldane compound by mean of state of the art of Resonant Inelastic X-ray Scattering (RIXS) experiments. The high quality data presented in the manuscript are interpreted in terms of an isotropic Heisenberg model, whose dynamical structure factor is calculated by mean of the density matrix renormalization group (DMRG) technique. The authors provide a careful characterization of the spectral features corresponding to the quadrupolar ΔS_2 excitations by measuring their variation as a function of momentum transfer and temperature. In this way they unveil the presence of both coherently and incoherently propagating triplet pairs in the system. I find the present manuscript properly written and well organized. The quality of the data is impressive and represents the state of the art of RIXS experiments. I find the interpretation of the spectral features corresponding to quadrupolar excitations and the comparison with the theoretical model convincing. However, I do not find surprising that RIXS can probe quadrupolar excitations. As presented by A. Nag *et al.* in Ref. 21 single site quadrupolar excitations were already measured in NiO. More interesting in my opinion is the dynamics of such excitations revealed in a compound such the $S = 1$ Haldane chain material as Y_2BaNiO_5 . The importance of such a result with respect to open problems in magnetism and, more generally, strongly correlated systems is not stressed enough in the present manuscript. In my opinion, in order to meet the broad interest required by Nature communications the authors must state more clearly and more precisely the general interest of their results.

Our Response: We thank the Reviewer for their time and effort in reviewing our manuscript and for encouraging us in stressing more the the broad interest of our work. Following their recommendations, we have now improved the discussion about this in the *Outlook* section of the main text and renamed it now to *Discussion and Outlook*. Here, we provide an overview of the importance and broad applicability of our work.

In quantum materials, the higher-order multipolar spin excitations are usually weaker than the dipolar spin counterparts but give rise to exotic non-classical phenomena *in a wide variety of magnetic systems* like Kitaev spin liquids, spin-nematicity, on-site multiferroicity, and bound-magnon states, etc. Such excitations are also proposed to play a role in the iron pnictide superconductors¹. Similarly bilinear biquadratic exchanges play an important role in the recently discovered van der Waals magnets. Unfortunately, the multipolar excitations are challenging to study using conventional probes such as inelastic neutron scattering which is limited by dipolar selection rules. In rare situations, multipolar spin excitations were successfully observed, where they are perturbed by anisotropic spin-exchange, spin-orbit coupling, or large magnetic fields². Therefore, establishing an experimental probe capable of probing directly the *dynamics* of multipolar spin excitations in quantum materials is crucial.

In our previous work, (Ref. 21 in manuscript) we indeed show that single site quadrupolar excitations can be probed using RIXS. However, we could not demonstrate whether the

¹See, for example, the cited references 11 in the manuscript.

²See, for example, the cited references 12-17 in the manuscript.

quadrupolar excitation observed by RIXS is only a local double spin-flip or it can also probe the dynamics of such excitations in momentum space. *Here, we have shown that the quadrupolar excitations disperse and hence RIXS is not just observing a local double spin-flip and therefore important for unveiling multipolar dynamical properties.* For example, there was a considerable period of time, when it was thought that RIXS could only observe local dipolar spin flip or two site bimagnon excitation until it was convincingly demonstrated that it could probe magnon dispersions³.

Focusing on the paradigmatic example of spin-1 chain Y_2BaNiO_5 , we have demonstrated that the pure quadrupolar excitations in an isotropic Haldane system, without invoking anisotropic interactions, can be observed. Recently, in spin-1/2 Heisenberg antiferromagnetic spin chains, it has been seen that Cu L -edge RIXS can probe both spin-conserved and non-spin-conserved higher-order four-spinon excitations⁴. Our present work further consolidates L -edge RIXS, as a powerful probe for characterising non-local long-range magnetic correlations via the study of higher-order spin-flip excitations, thereby, extending both the energy-momentum phase space and the diversity of the magnetic quantum systems that can be explored. We have specifically chosen the Haldane system for its isotropic nature, but our results are broadly applicable to any magnetic system in higher dimensions.

Specifically, by studying the spin-1 Haldane chain Y_2BaNiO_5 , we discovered that quadrupolar excitations have an unusual spectral weight distribution, featuring a low energy continuum and a dispersive antibound state above the two particle continuum predicted by the field theory. In this regard, it is well known that systems with symmetry protected topological (SPT) order such as the one investigated in our work are characterized by gapped bulk excitations and non-trivial fractionalized gapless edge states. Our work explores experimentally for the first time bulk multipolar excitations in a SPT phase, and it shows that these excitations exhibit unusual fingerprints that might help detect and recognize SPT phases in other strongly correlated materials.

³Braicovich L. et al. Magnetic Excitations and Phase Separation in the Underdoped $\text{La}_{2-x}\text{Sr}_x\text{CuO}_4$ Superconductor Measured by Resonant Inelastic X-Ray Scattering Phys. Rev. Lett. 104, 077002 (2010).

⁴Kumar, U. et al. Unraveling higher-order corrections in the spin dynamics of rixs spectra (2021). arXiv: 2110.03186

Report of the Third Referee

The Referee wrote: Excitations in magnetic materials that involve a single spin-flip are easy to observe by standard methods, like neutron scattering or ESR. These are well-known magnons (sometimes triplons), and their dispersion provides valuable information on the properties of the phase and on the strength of interactions. They are also the Goldstone-modes of the ordered magnets with broken time-reversal symmetry. Magnetic systems in the quantum limit may realize phases that are beyond the usual picture of ordered magnetic moments. The authors present an experimental and numerical study of the quadrupolar excitations in a quasi-one-dimensional material Y_2BaNiO_5 . In this material, the $S = 1$ spins of the Ni ions form a quantum AKLT-state with a singlet ground state separated by a gap from the excitations. The resonant inelastic X-ray scattering (RIXS) excites two spin-flip modes - these are the so-called quadrupolar excitations. They are extremely difficult to create and observe, as the experimental probes usually do not interact with them, with the exception of the light. In the actual experiment, the two-spin quadrupolar excitations decay into a continuum of two triplet excitations and a quadrupolar collective mode at higher energies, which is an anti-bound state of two triplets. The authors compare in detail the RIXS spectrum with numerical calculation of the $S = 1$ spin chain and identify the origin of the different parts of the spectrum. In my opinion, this is very nice work and I would like to recommend the acceptance of the manuscript for publication. The presented experimental technique may become essential to provide experimental identification of nematic (quadrupolar) ordering of spins, suggested for some materials, by observing their Goldstone modes.

However, before publication I would like to make a few suggestions, mostly esthetic ones:

Our response: We thank the Reviewer for their time and efforts reviewing our manuscript, their positive recommendation, and their helpful comments. We agree with all of them and have addressed each as detailed below.

The Referee wrote: (1) The process depicted in Fig. 1(b) is essential. However, it is very difficult to comprehend what is precisely happening. I think the initial state is missing - would it be two spins on $3d$ orbitals and 2 spins on $2p$ orbitals? And I am confused with the number of white and colored arrows. Could the authors spend some time and improve the figure?

Our response: We have revised Fig. 1b and included the initial and final states in the RIXS process following the suggestion by the referee. We have also added a sentence in the figure caption ‘*The coloured (white) arrows represent the occupied (empty) spin states of the electrons involved in the process.*’ to clarify the different colours of the electrons.

The Referee wrote: (2) Fig. 2 shows the measured spectra. The ticks are difficult to discern. Could they color ticks differently (say green?). Also, I was trying to extend the figure to $q_{\parallel} = 0.5$, to the zone boundary. I think it would help to show where is the boundary of the (1D) Brillouin zone, and to show the fitted dispersion (dashed line) in the whole zone. Clearly, there will be some white space where there is no RIXS signal, but that would also be informative.

Our response: Following the suggestion of the referee, we have now coloured the ticks in green and extended the maps to $q_{\parallel} = 0.5$ with the fitted dispersion shown over the whole zone.

The Referee wrote: (3) I am confused with the sentence (line 131 of the manuscript): “because the quadrupolar excitations can only occur at the $S_z = \pm 1$ sites and not the $S_z = 0$ sites.” Just like magnons can be excited with any of the S^+ , S^z , and S^- operators, there is a $(S^z)^2 - 2/3$ quadrupolar operator which will act on the $S_z = 0$ sites (there are five quadrupolar operators, as they are rank-2 tensor operators, see e.g. the chapter about spin operators in the Messiah’s textbook on Quantum Mechanics). Could the authors improve that part of the text (I understand that there is a conflict between simplicity and exactness, but as the sentence now reads, it may be misleading)

Our response: We thank the Referee for this observation. We in fact completely agree that the sentence highlighted above was misleading. We have modified it to now read

The ΔS_2 excitations also have an exponentially decaying correlation length, albeit smaller than ΔS_1 excitations. The difference in correlation length is likely due to the fact that the creation of quadrupolar wave excitations costs energetically at least twice as much the single spin-flip magnon-like excitations at any wave-vector.

This formulation highlights that, as *single* spin excitations, the quadrupolar excitations in spin $S = 1$ chains are also gapped at any wave-vector with at least twice energy cost.

The Referee wrote: (4) I think the Supplemental Material will be printed as it is provided. If I am not mistaken, all the formulas were typeset using Word. It was really painful to read them - how about converting those parts of the text to latex (or anything that looks reasonable)?

Our response: Both the main text and supplement were typeset in L^AT_EX. We had used a Nature journal style file, which rendered all of the fonts including the equations in a sans-serif font. We have now changed the relevant style a standard RevTex4-2 format that follows typical L^AT_EX documents.

List of changes

1. Upon the first Reviewer's request, we have expanded our discussion to outline the importance of our work for the boarder research community.
2. Following the third Reviewer's suggestion, we have modified Fig. 1 of the main text to clarify the initial, intermediate, and final atomic states involved in the scattering process.
3. We expanded the experimental data plots in Fig. 2 to cover the entire first Brillouin zone.
4. We have revised the discussion of the action of the different quadrupolar operators that appeared around line 131 of the original manuscript.
5. We have reformatted the supplement as per the third Reviewer's request.
6. We renamed the *Outlook* section to *Discussion and Outlook* to reflect the revised content.
7. We have corrected a few additional typos we identified while editing the manuscript.

Reviewers' Comments:

Reviewer #1:

Remarks to the Author:

I find that in the revised version of the manuscript "Quadrupolar magnetic excitations in an isotropic spin-1 antiferromagnet" by A. Nag et al. the authors better show the general interest of their result and the importance of using L-edge RIXS for revealing the dynamics of higher order magnetic correlations. This information is valuable for a broad community studying novel properties of quantum materials, in which magnetic correlations play a key role. For this reason I support for publication in Nature Communications this revisited version of the manuscript.

Reviewer #3:

Remarks to the Author:

In the resubmitted manuscript, the authors addressed my comments from the first review and improved the figure. I would like to recommend the manuscript for publication.

Report of the First Referee:

The referee wrote:

I find that in the revised version of the manuscript "Quadrupolar magnetic excitations in an isotropic spin-1 antiferromagnet" by A. Nag et al. the authors better show the general interest of their result and the importance of using L-edge RIXS for revealing the dynamics of higher order magnetic correlations. **This information is valuable for a broad community studying novel properties of quantum materials, in which magnetic correlations play a key role. For this reason I support for publication in Nature Communications this revisited version of the manuscript.**

Our response:

We thank the referee for suggestions on this point during the first review. We are glad that the referee supports publication of our present version in Nature Communication.

Report of the Second Referee:

The referee wrote:

In the resubmitted manuscript, the authors addressed my comments from the first review and improved the figure. **I would like to recommend the manuscript for publication.**

Our response:

We thank the referee for suggesting us to improve Figure 1 of the main article. We are glad that the referee supports publication of our present version in Nature Communication.